# Environmental Impact Analysis of Alkali-Activated Concrete with Fiber Reinforcement

**Pujitha Ganapathi Chottemada** [1,*] **, Arkamitra Kar** [1] **and Patricia Kara De Maeijer** [2]

[1] Department of Civil Engineering, Birla Institute of Technology and Science—Pilani, Hyderabad Campus, Hyderabad 500078, Telangana, India; arkamitra.kar@hyderabad.bits-pilani.ac.in

[2] EMIB, Faculty of Applied Engineering, University of Antwerp, Groenenborgerlaan 171, 2020 Antwerp, Belgium; patricija.karademaeijer@uantwerpen.be

* Correspondence: p20200432@hyderabad.bits-pilani.ac.in

**Abstract:** The scientific community is shifting its focus towards construction materials with a low carbon footprint, such as alkali-activated concrete (AAC). The present study conducts an environmental impact assessment using the cradle-to-grave approach to examine the environmental implications of three different types of ambient-cured AACs with varied combinations of precursors. The 28-day compressive strength values of the concrete mixes used for comparing environmental impacts vary from 35 to 55 MPa. Among these mixtures, the one with the least environmental impact is chosen for further impact assessment with the inclusion of fibers. Three different fiber reinforced AAC mixes containing 0.3% steel, glass, and polypropylene fibers, respectively by volume of AAC, are used in this study. The findings show that Portland Cement concrete has 86% and 34% higher impacts than AAC on the ecosystem and human health, respectively. In the production of AAC, sodium silicate is found to have the highest impact on the environment, in the range of 30–50% of the total impact. Among the various fibers used, glass fibers have the highest impact, which is 12% and 13% higher than that of the plain AAC mix, on the ecosystem and human health, respectively.

**Keywords:** life cycle assessment; alkali-activated concrete; life cycle impact assessment; fiber reinforced alkali-activated concrete





## 1. Introduction

An enormous amount of energy is used in the construction industry, from the mining and processing of raw ingredients to the manufacture and installation phases [1]. With an eye towards decarbonization, the World Green Building Council (WGBC) has established an aim to reduce the embodied $CO_2$ emissions of all upcoming construction, infrastructure, and renovation projects by at least 40% by the end of 2030 and to reach net zero by 2050 [2]. It has also been widely recognized that the demolition of buildings and the disposal of building materials at the end of their useful lives are major sources of environmental damage [3]. As a result, research and development initiatives have been focusing on creating energy efficient materials that can reduce the impacts of construction on the environment without sacrificing the performance requirements of conventional building materials. However, it is crucial to quantify the repercussions of innovative technologies in infrastructure in terms of both economic and environmental aspects in order to support and promote their implementation. Hence, the primary focus of this study is to conduct life cycle assessments (LCA) for different concrete systems to identify their relative environmental impacts and to provide recommendations for practical implementations.

The sustainability of concrete and the construction industry is being enhanced by the constant evolution of Portland Cement (PC) substitutes. Among these alternatives, alkali-activated concrete (AAC) has gained substantial scientific interest. AAC was first introduced by Davidovits in 1979 [4] as an inorganic polymer obtained by the alkali-activation of aluminosilicate-rich precursors. The most frequently used precursors for the

production of AAC are industrial residues and by-products such as fly ash and ground granulated blast furnace slag (GGBS), making it an environmentally sustainable alternative. AACs made of fly ash and GGBS have undergone extensive research, and the results indicate that their mechanical and durability performance is close to or even greater than that of PC concrete [5–7]. Though the suitability of AAC to replace PC concrete has been established in terms of strength and durability aspects, there are very few studies focusing on the environmental impacts of AAC [8–15]. Existing studies have indicated that, while AAC has a substantially smaller impact on $CO_2$ emissions compared to PC concrete, the alkaline-activators used in the production of AAC have negative impacts on other environmental aspects [16,17]. However, replacing fly ash precursor with GGBS permits AAC to be cured at room temperature and requires a lower amount of alkaline-activator to generate comparable mechanical strength [18]. This, in turn, can help mitigate the harmful effects of the alkaline-activator.

Furthermore, on account of their superior mechanical properties and crack bridging effect, fiber reinforcements have also been widely used in construction activities [19–22]. However, the manufacture and use of synthetic fibers in concrete are associated with high economic costs, significant environmental impact, and increased energy demand [23]. On the contrary, natural fibers have been found to have a number of environmental benefits, including low pollutant emission and zero carbon footprint, as well as lower production costs and energy consumption [24–26]. Nevertheless, a major drawback with employing natural fibers as reinforcement for concrete is their poor mechanical performance, which is attributed to the alkaline environment that can be detrimental to natural fibers [27]. On the other hand, the metal fibers incorporated in AAC have demonstrated greater mechanical strength, owing to AAC's high stiffness, unlike synthetic and natural fibers with a low modulus of elasticity [28]. An existing study [29] reported that even small proportions of polypropylene and steel fibers in AAC result in a significant reduction in shrinkage. Though there are numerous studies [30–34] on the mechanical performance of fiber reinforced AAC (FRAAC), there are limited studies [13,35,36] pertaining to the environmental sustainability of incorporating these fibers in AAC.

Evaluating the eco-efficiency of concrete by analyzing its impact on the environment is a complex process that calls for a wide range of techniques. LCA is one such assessment method that can be employed to evaluate the potential impacts and risks associated with a material from its "cradle-to-grave", i.e., throughout the life of a product system from the acquisition of resources to the manufacture, utilization, and disposal [37]. In recent years, several LCAs have been conducted to compare the environmental effects of PC concrete with AAC. The findings indicate that PC concrete leads to global warming potential (GWP) that is twice that of AAC [9]. However, the use a traditional alkaline-activator (sodium hydroxide + sodium silicate) in AAC significantly affects the environment. According to previous research [8], sodium silicate in AAC contributes about 50–59% to global warming and 41–53% to terrestrial acidification, terrestrial ecotoxicity, and photochemical oxidant formation. This is attributed to the melting and vitrification phases that take place in furnaces at extremely high temperatures and pressures during the manufacturing of sodium silicate [36]. Employing various precursors in alkali-activated systems also produces a varying range of outcomes. Adopting high-volume fly ash in alkali-activated mortars emits 90% less $CO_2$ than PC mortar, whereas palm oil fly ash and slag at high volumes emit 45% and 84% less $CO_2$, respectively [38]. Although alkali-activated concrete is effective in reducing greenhouse gas emissions and GWP, it may not always have a smaller impact in other areas, such as ecotoxicity and ozone depletion. This is primarily due to the usage of alkaline-activators [14,16]. However, the emissions were reduced by 60–99% [13,15] when rice husk ash, waste glass, and desulfurization dust were employed in place of these traditional activators. An existing study [36] employing glass, basalt, and hemp fibers in alkali activated mortars revealed that the addition of 1% by weight of these fibers generates an average of 38 $kgCO_2$-eq of $CO_2$ emissions. There were no significant variations in $CO_2$ emissions resulting from the use of naturally or synthetically derived

fibers. When compared to conventional PC concrete, LCA on AACs employing glass, steel, and polypropylene fibers demonstrated reduced emissions of 23%, 16%, and 0.8%, respectively, and decreased emissions of 37%, 32%, and 19% when compared to steel fiber reinforced PC concrete [13]. However, as a result of variations in the mix design of each FRAAC mixture taken into account to reach comparable strength values, these results do not directly correlate with the type of fiber used. An environmental analysis of PVA fiber-reinforced alkali-activated slag foam concrete revealed that 60% of abiotic resource depletion potential was caused by sodium silicate, followed by PVA fibers, with 22% impact [35].

LCA is a great tool for guiding the selection of an alternative building material [10,11,39,40] that has the least negative impacts on the environment. Although studies on various AAC and FRAAC mixtures have been conducted in the past, it is still unclear how different precursors and fibers utilized may have an impact on the environment. Hence, the current study considers a consistent mix design, with the type of fiber and binder being the only two variables, and exclusively focuses on the environmental impacts of the binder and fibers employed. Therefore, the present study helps validate the practical use of AAC and FRAAC by assessing the life cycle impacts involved in its production. The specific objectives of the present study are (i) to select three different ambient-cured AAC mixtures of comparable strength to identify the optimal precursor combination using LCA; (ii) to evaluate the impacts of FRAACs made of three different types of fibers: steel, glass, and polypropylene, on the environment; and (iii) to assess the economic feasibility of FRAAC using a simplified cost analysis.

## 2. Materials and Methods

### 2.1. Materials

Class F fly ash and GGBS are considered as precursors for AAC in the present study. Fly ash is a fine-grained residue of coal combustion, typically produced in thermal power plants. Class F fly ash contains pozzolanic components such as glassy silica and alumina which, in the presence of a cementing agent such as PC, quicklime, or hydrated lime and water, readily reacts to generate cementitious compounds. Alternatively, adding an alkaline-activator to fly ash, such as sodium silicate, can lead to the formation of a geopolymer [41]. As a result of the high concentration of coal-fired power stations, the Asia–Pacific region holds the greatest share of the global market for fly ash. In 2021, China accounted for more than 50% of the global fly ash market. Due to its vast population and expanding infrastructural requirements, India is another key producer and consumer on the fly ash market. More than 285 coal-fired power stations produce around 500 million tons of fly ash annually in India. Nevertheless, 20.82% of this fly ash remains unutilized, according to previous reports [42]. Fly ash for the present study is supplied by the National Thermal Power Corporation plant located in Ramagundam, Telangana, India.

Slag is a by-product in the production of pig iron that is obtained in a molten state. Granulated slag, which has inherent hydraulic properties, is formed when molten slag is rapidly quenched with water that leads to the formation of a fine, granular, nearly non-crystalline, glassy component. When blended with PC, such granulated slag has been demonstrated to have good cementitious properties [43]. The worldwide pig iron production increased in 2018 to about 1247 million tons (Mta) from 1212 Mta in 2017, resulting in the production of about 330 to 375 Mta of blast furnace slag [44]. AAC produced with fly ash alone as the precursor necessitates elevated curing regimes to achieve better strength and faster geopolymerization [45]. It has been reported that the addition of GGBS to fly ash in the production of AAC gives a reasonably high compressive strength under ambient curing conditions [41,42]. This is attributed to the C–A–S–H (calcium–aluminum–silicate–hydrate) and C–S–H (calcium–silicate–hydrate) matrix formed in addition to N–A–S–H (sodium–aluminum–silicate hydrate), which forms with low calcium fly ash alone [46,47]. For the present study, GGBS is acquired from ASTRAA chemicals, Chennai, India.

The present study also uses PC as the binder to produce conventional concrete. Ordinary Portland Cement (Type I) complying with the grade 53, as per [48,49], is used. The properties of the various precursors used in the present study are provided in Table 1.

**Table 1.** Chemical composition of binders *.

| Composition (%) | GGBS | Fly Ash | PC |
|---|---|---|---|
| CaO (%) | 37.63 | 3.80 | 65.23 |
| $SiO_2$ (%) | 34.81 | 48.81 | 18.64 |
| $Al_2O_3$ (%) | 17.92 | 31.40 | 5.72 |
| MgO (%) | 7.80 | 0.70 | 0.85 |
| $SO_3$ (%) | 0.20 | 0.91 | 2.34 |
| $Fe_2O_3$ (%) | 0.66 | 7.85 | 4.54 |
| $TiO_2$ (%) | - | 2.93 | 0.5 |
| $K_2O$ (%) | - | 1.52 | 0.59 |
| $Na_2O$ (%) | - | 1.04 | - |
| MnO (%) | 0.21 | - | - |
| LOI (%) | 1.41 | 3.00 | 1.69 |
| Strength activity index (%) | 114.46 | 96.46 | - |
| $d_{50}$ (μm) | 13.93 | 51.90 | - |
| Blaine fineness ($m^2$/kg) | 386.00 | 327.00 | 285 |
| Specific gravity | 2.71 | 2.06 | - |

* Supplied by manufacturer.

Glass fibers are primarily made of $(SiO_2)_n$ monomers found in naturally formed mineral deposits. The production of these fibers involves the further extrusion of molten parent material into the form of a filament [31]. The use of glass fibers in fly ash-based AAC has been the subject of several research studies [50–53]. It is reported that an optimal dosage of 0.3% by weight of glass fibers [51] is required to produce FRAAC with a compressive strength of 24.8 MPa. On the other hand, the inclusion of 0.3% glass fibers by volume resulted in a compressive strength of 31.7 MPa, split tensile strength of 2.69 MPa, and flexural strength of 3.28 MPa, increasing the strength by 3.26%, 31.57%, and 52.37% compared to AAC without fibers [34,54]. However, a significant reduction in slump was observed with the increase in fiber addition. Similar research [54] revealed an optimal increase in compressive, split tensile, and flexural strength of 24.8%, 25.5%, and 12.81%, respectively, with the inclusion of 0.5% glass fibers. The glass fibers considered for the present study have a specific gravity of 2.6.

Steel fibers are extensively utilized in concrete mixtures due to their enhanced mechanical strength, adaptability, and availability [55]. These fibers may be either industrially produced or wastes fibers generated in industries. Recent studies have demonstrated significant strength gains with the addition of waste lathe fibers [56] and waste tire steel fibers in concrete [57]. An existing study [58] revealed that the compressive strength of AAC at 28 days increased by 2.1% and 3.66%, respectively, with the inclusion of steel fibers at 0.3% and 0.6%. However, the splitting tensile strength at 28 days increased by 1.4% with the inclusion of 0.3% steel fibers, while it decreased by 1.95% with the addition of 0.6% steel fibers. Therefore, with a compressive strength of 44.18 MPa and a split tensile strength of 3.64 MPa, AAC with 0.3% steel fibers by volume was shown to be optimal. Several studies demonstrate the beneficial effects of steel fiber inclusion on the flexural strength and splitting tension for both fly ash and slag-based AACs. However, with the addition of steel fibers, there is a minimal gain [58–60] or a negative impact [61,62] on compressive strength. The present study considers industrially produced steel fibers, and a specific gravity of the steel fibers employed for this investigation is 7.85.

Polymer fibers, such as polypropylene, are characterized as straight or distorted pieces of extruded, aligned, and chopped polymer material [63]. Theses fibers are manufactured at high temperatures and have high chemical resistance, alkaline stability, and are hydrophobic in nature. However, owing to its low modulus of elasticity, barring few exceptions [64,65], its incorporation has mostly shown a decreasing effect on the compressive

strength of AACs [58,66–71]. Adding a small dosage (0.1–0.3%) of polypropylene fibers to AAC improves the microstructure and inhibits the development of microcracks [72]. The addition of 0.3% and 0.6% polypropylene fibers to slag-based AAC resulted in a compressive strength of 54.4 MPa and 53.23 MPa, respectively [58]. It has been demonstrated that an increased dosage of polypropylene fibers has a favorable impact on flexural strength, mostly due to its ability to bridge cracks [28]. However, with a fiber dosage above 0.3% volume fraction, it begins to decline. This decrease is mostly due to the poor compaction of the high fiber composite, which ultimately leads to the formation of a heterogeneous porous structure. Furthermore, synthetic micro-fibers such as polypropylene and PVA have been proved to significantly reduce drying shrinkage when used at 0.1% volume [73]. The specific gravity of the polypropylene fibers taken into consideration for the present investigation is 0.905.

According to previous research [28,58], the inclusion of fibers in AAC beyond 0.3% volume fraction mostly leads to poor compaction and a heterogeneous mixture, thereby compromising the mechanical properties of FRAACs. Hence, the present study adopts FRAAC mixes containing a fiber dosage of 0.3% by volume to perform LCA. The images of raw materials and sample test specimens are presented in Figure 1.

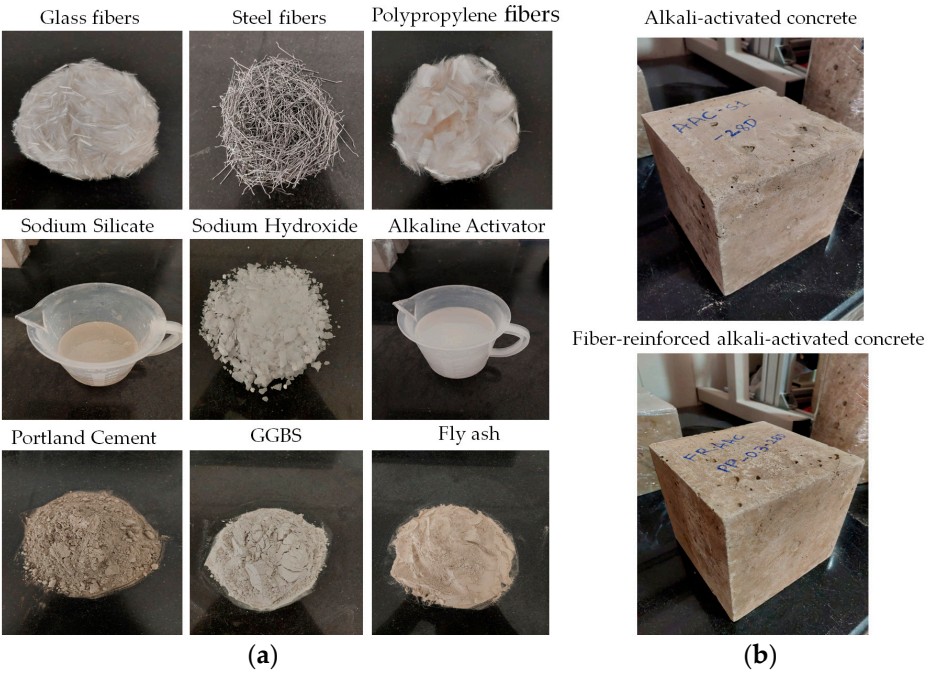

**Figure 1.** (**a**) Raw materials and (**b**) sample test specimens.

### 2.2. Methodology

The mix proportions selected in this study comply with the requirements of the S4 class of slump as mentioned in BS EN 12350-2 [74]. The combinations of fly ash and GGBS as precursors and the corresponding alkali activators are presented in Table 2. As mentioned earlier, the dosage of glass, steel, and polypropylene fibers is maintained at 0.3% by volume of AAC to prepare the FRAAC mixtures. The compressive strength values of the mixes used in the study vary from 35 to 55 MPa at the age of 28 days.

For this study, LCA is first conducted on the plain AAC and PC concrete mixtures to evaluate the environmental impacts of the binders. The mixture with the least environmental impacts is then chosen for further LCA on FRAACs. The results are then analyzed at both midpoint and endpoint categories, followed by a simplified cost analysis to assess the economic feasibility of both the plain and fiber-reinforced concrete mixtures. The flow of research is presented in Figure 2.

**Table 2.** Mix compositions (kg/m³) *.

| Material | S100 | FS70 | FS50 | PCC | FS50—SF0.3 | FS50—GF0.3 | FS50—PP0.3 |
|---|---|---|---|---|---|---|---|
| GGBS | 450 | 315 | 225 | - | 225 | 225 | 225 |
| Fly ash | - | 135 | 225 | - | 225 | 225 | 225 |
| Cement | - | - | - | 450 | - | - | - |
| Sodium silicate | 65.82 | 65.82 | 65.82 | - | 65.82 | 65.82 | 65.82 |
| Sodium hydroxide | 12.48 | 12.48 | 12.48 | - | 12.48 | 12.48 | 12.48 |
| Water | 170.21 | 170.21 | 170.21 | 150 | 170.21 | 170.21 | 170.21 |
| Fine aggregates | 645 | 645 | 645 | 623 | 645 | 645 | 645 |
| Coarse aggregates | 967 | 967 | 967 | 1084 | 967 | 967 | 967 |
| Steel fibers | - | - | - | - | 23.55 | - | - |
| Glass fibers | - | - | - | - | - | 7.80 | - |
| Polypropylene fibers | - | - | - | - | - | - | 2.715 |

* All values are in kg/m³; PC—Portland Cement; S—Ground granulated blast furnace slag; F—Fly ash; SF—Steel fibers; GF—Glass fibers; PP—Polypropylene fibers. The numerical values in the mix nomenclature indicate the proportion of the preceding ingredient.

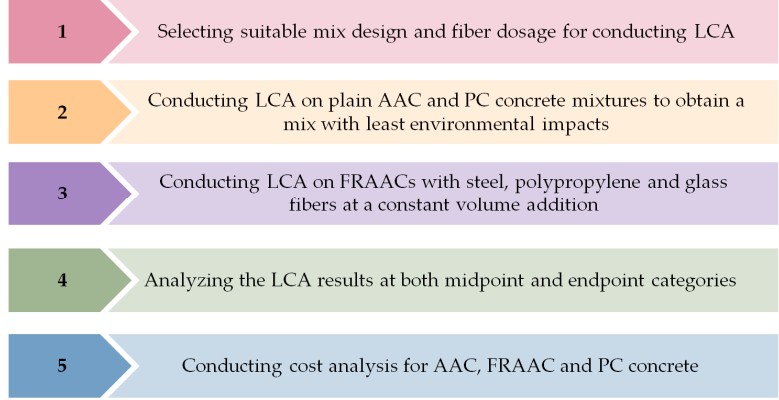

**Figure 2.** Research flow.

Concrete Mix Design Description

The present study considers three different mix designs for AAC and one for PC concrete. A consistent mix design has been maintained for all AAC mixtures, with the precursors being the only variable. Sodium hydroxide and sodium silicate mixed with a constant activator modulus of 0.87 is adopted as alkaline-activator in all AAC blends. These mix proportions are designed with an aim to produce ambient-cured AACs of comparable strength. The mix compositions are presented in Table 2. The mix S100 is a 100% GGBS-based mix, the mix FS70 consists of 30% fly ash and 70% GGBS as the precursor, mix FS50 consists of 50% fly ash and 50% GGBS, and the mix PCC contains PC as the binder. The most environmentally sustainable mixture is then selected based on life cycle impact assessments for further analysis on FRAAC. Three distinct fibers, namely, steel, polypropylene, and glass fibers, are added at a constant volume of 0.3% for further life cycle impact analysis. The mix FS50–SF0.3 contains 0.3% of steel fibers in the FS50 mix, FS50–GF0.3 contains 0.3% of glass fibers in the FS50 mix, and FS50–PP0.3 consists of 0.3% polypropylene fibers in the FS50 mix.

### 2.3. Life Cycle Assessment Method

LCA is a systematic method for comparing the environmental and energy efficiency of building materials derived from various raw materials for the same application [75]. The four stages of an LCA study, as outlined by ISO standards [37,76], are as follows: (i) defining the objective and scope of the study, (ii) conducting a life cycle inventory analysis, (iii) conducting a life cycle impact assessment, and (iv) analyzing the outcomes.

2.3.1. Objective and Scope

The primary objective of this research is to investigate the effects of various combinations of fly ash and GGBS as binders in AAC, on the surrounding ecosystem, as well as to assess the effect of various fiber additions in AAC on the environment. The results are subsequently analyzed to determine the effectiveness of the precursors and fibers in enhancing the environmental sustainability of AAC and FRAAC. The economic feasibility of these blends is also evaluated through a cost analysis that takes into account the current situation in India. All assessments are carried out using 1 m$^3$ of concrete as the functional unit for both environmental and economic analyses.

The present study evaluates the environmental impacts of four different mix proportions, namely: (i) AAC with GGBS as the sole precursor (S100), (ii) AAC with 30% fly ash and 70% GGBS as precursor (FS70), (iii) AAC with 50% fly ash and 50% GGBS as precursor (FS50), and (iv) conventional concrete with PC as the binder. Based on the LCA results, the most environmentally sustainable mixture is then selected as the optimal mix for further analysis on FRAAC. Three distinct fibers, namely steel, polypropylene, and glass fibers, are added at 0.3% volume fraction to the optimal mix in order to evaluate their environmental sustainability. The LCA technique adopted uses the cradle-to-gate approach, which includes estimates of all emissions and energy use from the acquisition of raw materials to the manufacture of concrete. Since the primary aim of this study is to compare the environmental impacts of various AAC mixtures with ordinary PC concrete, the environmental impacts of the utility and disposal phases are assumed to be equal.

2.3.2. Inventory Analysis

The life cycle inventory (LCI) phase gathers and processes the inputs and outputs required to create a product system [37]. Since LCA is a system study, it is crucial to understand the idea of system boundaries. In general, the boundary between the product and the environment is established by the representation of the complete life cycle of a product as a technical system in LCA. Therefore, the supplies to all upstream processes should ideally be traced back to resources as they are found in nature [77]. Finding the appropriate processes in the product system and acquiring the database for subsequent quantification of inputs and outputs for the intended functional unit are the two main challenges in the inventory analysis. Ecoinvent 3.7.1 [78] provides the LCI data for the processes involved in the current investigation. Since there is no direct process available for steel and polypropylene fibers in the Ecoinvent database, wire drawing steel and non-woven polypropylene are employed, respectively, as the unit processes for these fibers. The process of wire drawing employs steel wire rods of diameters ranging from 5.5 to 16 mm from hot rolling mills and reduces them to thin wires with a diameter of about 1–2 mm. The process of non-woven polypropylene involves the use of polypropylene granulates to produce non-woven spun bond polypropylene. These processes are adopted as they match closely with the actual production of steel and polypropylene fibers. Figure 3 shows the system boundary of FRAAC for the cradle-to-gate LCA considered in this research.

Furthermore, the methodological guidelines used to calculate the database are specified by the system models. To ascertain the supply and distribution of impacts between producers and consumers of goods and services, all system models begin with the same pool of discrete human activity (allocation and substitution). Allocation at the Point of Substitution (APOS) is chosen as the system model for the current investigation. It is based on an attributional approach wherein the burden of producing wastes is shared between waste generators and individuals who ultimately reap the economic and practical benefits of waste treatment [78].

The transportation and unit costs of the raw materials in Table 3 are presented in Indian Rupees (INR) for the Indian scenario. The costs listed in the table for each raw material include the freight charges. The freight distances represent the distance between the manufacturing facility and the BITS—Pilani, Hyderabad Campus.

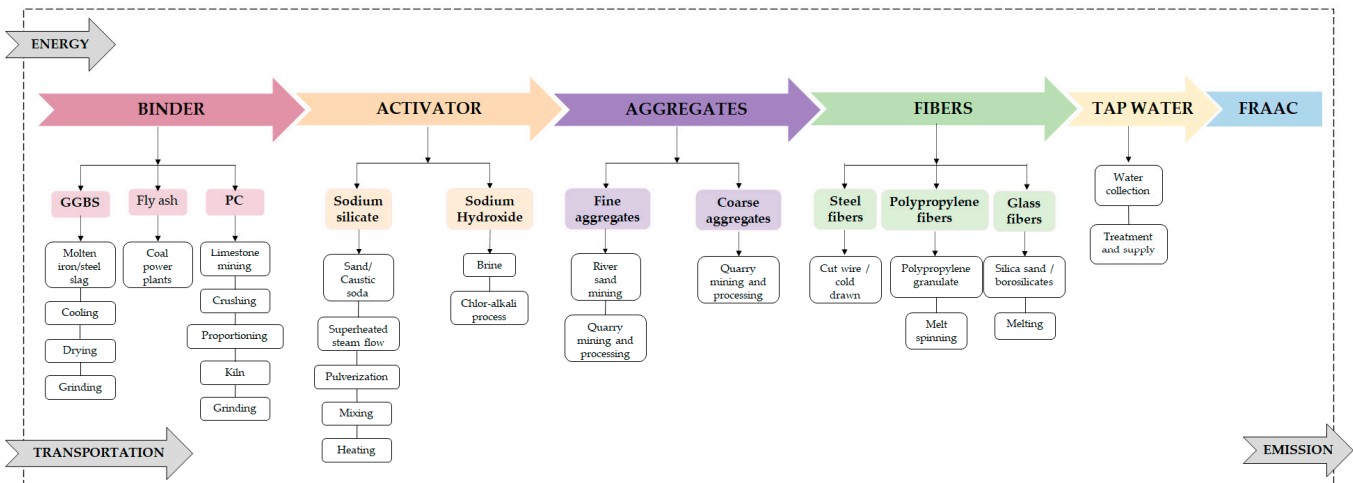

**Figure 3.** System boundary of FRAAC for cradle-to-gate LCA.

**Table 3.** Unit cost and freight distance of raw materials.

| Raw Materials | Distance (km) | Cost (per kg in INR) |
|---|---|---|
| Fly ash | 202 | 4.8 |
| GGBS | 648 | 18 |
| Portland Cement | 231 | 8.3 |
| Sodium hydroxide | 28.4 | 106.2 |
| Sodium silicate | 22.8 | 81 |
| Fine aggregates | 14.6 | 2.6 |
| Coarse aggregates | 14.6 | 0.65 |
| Water | - | 0.035 |
| Glass fibers | 640 | 466.2 |
| Steel fibers | 23.4 | 156 |
| Polypropylene fibers | 23.4 | 494 |

### 2.3.3. Life Cycle Impact Assessment (LCIA)

Life cycle impact assessment (LCIA) is a phase of LCA that aims to comprehend and evaluate the magnitude and relevance of the possible impacts of a product system on the environment [37]. The present study uses ReCiPe 2016 midpoint and endpoint methodology to perform LCIA, as it provides a futuristic approach to transform a life cycle inventory to a number of life cycle impact scores at the midpoint and endpoint levels [79]. The primary focus of the ReCiPe approach is to create an updated method that integrates Eco-Indicator 99 and CML. Based on the global nature of many product life cycles, ReCiPe includes three endpoint categories and seventeen midpoint categories to provide characterization factors that are universally applicable. These indicators primarily highlight damage to human health, damage to ecosystems, and damage to resource availability [80]. The midpoint assessment contains indicators that are established midway between the emission and the endpoint, while the endpoint assessment contains indicators that are defined at the level of the areas of protection [77,81]. The current study adopts a hierarchical perspective, since a scientific consensus exists on both the period of time (100 years) and the validity of impact factors [82,83]. The open-source software, openLCA v 1.11.0, is used to estimate the impacts for each component of the system [84]. The impact categories listed in Table 4 are analyzed in this study with the ReCiPe method. Furthermore, to validate the reliability of the ReCiPe midpoint method, the study employs several other impact assessment methods, as suggested in the International Reference Life Cycle Data System (ILCD) handbook [82].

**Table 4.** Recipe midpoint impact categories.

| Midpoint Impact Category | Indicator | Characterization Factor | Unit |
|---|---|---|---|
| Climate change | Increase in infrared radiative force | Global Warming Potential (GWP) | kg $CO_2$-eq to air |
| Ozone Depletion | Stratospheric ozone decrease | Ozone Depletion Potential (ODP) | kg CFC-11-eq to air |
| Photochemical oxidant formation: terrestrial ecosystems | Increase in tropospheric ozone | Photochemical oxidant formation potential: ecosystems (EOFP) | kg NOx-eq to air |
| Terrestrial Acidification | Proton increase in natural soils | Terrestrial Acidification Potential (TAP) | kg $SO_2$-eq to air |
| Freshwater Eutrophication | Phosphorous increase in freshwater | Freshwater Eutrophication Potential (FEP) | kg P-eq to freshwater |
| Human Toxicity (Cancer) | Increase in risk of cancer disease | Human Toxicity Potential (HTPC) | kg 1,4 DCB-eq to urban air |
| Terrestrial Ecotoxicity | Hazard-weighted increase in natural soils | Terrestrial Ecotoxicity Potential (TEP) | kg 1,4 DCB-eq to industrial soil |
| Freshwater Ecotoxicity | Hazard-weighted increase in freshwater | Freshwater Ecotoxicity Potential (FETP) | kg 1,4 DCB-eq to freshwater |
| Marine Ecotoxicity | Hazard-weighted increase in marine water | Marine Ecotoxicity Potential (METP) | kg 1,4 DCB-eq to marine water |
| Fossil Resource Scarcity | Upper heating value | Fossil Fuel Potential (FFP) | kg oil-eq |

Note: Data extracted from [79].

2.3.4. Life Cycle Interpretation

The findings from the life cycle inventory and the impact assessment are integrated in a manner compatible with the established purpose and scope to provide recommendations and conclusion in the interpretation phase of LCA [37]. In the current study, the midpoint impact assessment results are presented by normalizing the total impact of each category to 100%. This helps in determining the material which bears the most negative impacts on the environment for each mix proportion. Further, the normalized scores for each of the midpoint categories are presented to assess the impact of each mix proportion on the various impact categories. Finally, the impact results for the endpoint assessment are presented for each area of protection to draw a conclusion on the most environmentally sustainable mix at the end of the cause–effect chain.

**3. Results and Discussion**

With AAC and PC concrete, the process contributions and the environmental impacts for ten midpoint impact categories are analyzed. The first section outlines the midpoint environmental impacts of four different plain concrete mixtures. Further, the endpoint assessment analyzes the impacts at the three areas of protection: damage to human health, quality of ecosystem, and scarcity of resources. Consequently, an environmentally sustainable mixture is selected and 3 different fibers are introduced into this optimal mixture at a constant volume fraction of 0.3% to evaluate the environmental impacts of the corresponding FRAAC mixture. A simple cost analysis is also conducted to assess the environmental feasibility of these concrete mixtures. As a result, the study recommends an environmentally friendly FRAAC mixture as a potential alternative to PC concrete.

*3.1. Plain Concrete*

3.1.1. Midpoint Assessment

The indicators provided by the midpoint characterization enable environmental impact assessment at a cause–effect loop between resource consumption towards the endpoint

level [77,81]. The LCIA results for ten distinct impact categories for plain concrete mixtures are presented in Figure 4. Normalized results are presented primarily to analyze the effect of individual material processes on the impact categories. It, thus, reveals which element in a specific product system contributes the most to detrimental effects on the environment. Table 5 provides the unit impact of each raw material on all the midpoint impact categories.

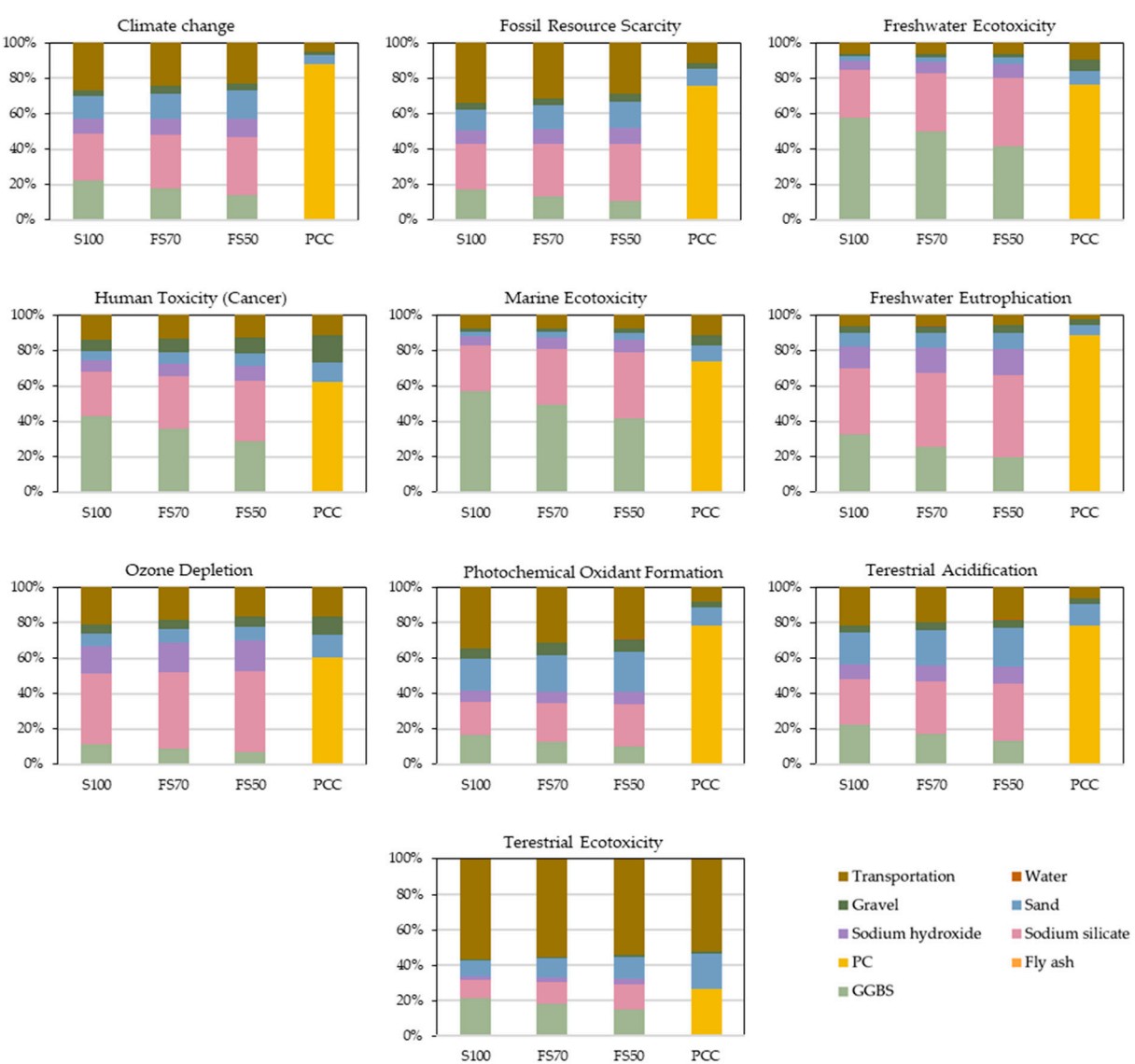

**Figure 4.** Process contribution (%) of AAC and PC concrete on midpoint impact categories.

**Table 5.** Unit impact of each material on the midpoint impact categories.

| Materials | GWP (kg CO$_2$-eq) | ODP (kg CFC-11-eq) | EOFP (kg NOx-eq) | TAP (kg SO$_2$-eq) | FEP (kg P-eq) | HTPC (kg 1,4 DCB-eq) | TEP (kg 1,4 DCB-eq) | FETP (kg 1,4 DCB-eq) | METP (kg 1,4 DCB-eq) | FFP (kg oil-eq) |
|---|---|---|---|---|---|---|---|---|---|---|
| Fly ash | 0 | 0 | 0 | 0 | 0 | 0 | 0 | 0 | 0 | 0 |
| GGBS | 0.10088 | $2.97 \times 10^{-8}$ | $2.60 \times 10^{-4}$ | 0.00036 | $4.70 \times 10^{-5}$ | 0.02181 | 0.58493 | 0.0213 | 0.02811 | 0.02076 |
| PC | 0.88534 | $8.10 \times 10^{-8}$ | $2.26 \times 10^{-3}$ | 0.0018 | 0.00016 | 0.01528 | 0.32095 | 0.00757 | 0.0102 | 0.11112 |
| Sodium silicate | 0.82784 | $6.98 \times 10^{-7}$ | $2.13 \times 10^{-3}$ | 0.00296 | 0.00037 | 0.0873 | 1.88792 | 0.067 | 0.08638 | 0.21411 |

**Table 5.** *Cont.*

| Materials | GWP (kg CO$_2$-eq) | ODP (kg CFC-11-eq) | EOFP (kg NOx-eq) | TAP (kg SO$_2$-eq) | FEP (kg P-eq) | HTPC (kg 1,4 DCB-eq) | TEP (kg 1,4 DCB-eq) | FETP (kg 1,4 DCB-eq) | METP (kg 1,4 DCB-eq) | FFP (kg oil-eq) |
|---|---|---|---|---|---|---|---|---|---|---|
| Sodium hydroxide | 1.31505 | $1.42 \times 10^{-6}$ | $3.49 \times 10^{-3}$ | 0.00479 | 0.00065 | 0.10876 | 2.11257 | 0.07081 | 0.09197 | 0.32831 |
| Sand | 0.04052 | $1.23 \times 10^{-8}$ | $2.10 \times 10^{-4}$ | 0.00021 | $7.49 \times 10^{-6}$ | 0.00192 | 0.16937 | 0.0006 | 0.0009 | 0.01002 |
| Gravel | 0.00736 | $5.73 \times 10^{-9}$ | $4.44 \times 10^{-5}$ | $3.06 \times 10^{-5}$ | $2.48 \times 10^{-6}$ | 0.00152 | 0.00754 | 0.00026 | 0.00034 | 0.00207 |
| Water | 0.00069 | $1.79 \times 10^{-10}$ | $1.59 \times 10^{-6}$ | $2.35 \times 10^{-6}$ | $3.47 \times 10^{-7}$ | $4.63 \times 10^{-5}$ | $2.19 \times 10^{-3}$ | $1.79 \times 10^{-5}$ | $2.40 \times 10^{-5}$ | $1.80 \times 10^{-4}$ |
| [a] PP | 3.67331 | $6.71 \times 10^{-7}$ | $8.40 \times 10^{-3}$ | 0.01054 | 0.00108 | 0.16727 | 2.75053 | 0.10024 | 0.13024 | 2.06367 |
| [b] SF | 0.33416 | $1.08 \times 10^{-7}$ | $5.60 \times 10^{-4}$ | 0.00074 | 0.00016 | 0.2654 | 0.31172 | 0.01734 | 0.02343 | 0.05525 |
| [c] GF | 2.56128 | $3.47 \times 10^{-6}$ | $9.91 \times 10^{-3}$ | 0.0126 | 0.00069 | 0.1393 | 3.72584 | 0.08608 | 0.11299 | 0.68652 |

[a] PP—Polypropylene fibers; [b] SF—Steel fibers; [c] GF—Glass fibers.

The GWP resulting from $CO_2$ production and greenhouse gas emissions is the most alarming impact category in the construction sector [85]. With nearly 90% contribution to GWP in the manufacturing of PC concrete, the findings clearly demonstrate that PC is the element that contributes the most to GWP. This serves as evidence of the significant carbon footprint caused by PC consumption [12]. Furthermore, sodium silicate used in the production of AAC is the second largest contributor to GWP after PC. The contribution of sodium silicate to GWP in AAC mixtures under consideration ranges from 25% to 35%. For AACs, the alkali-activators are the primary source for negative impacts in each category. This trend is in line with the previous investigation [8,12]. This is attributed to the high electricity and fuel consumption for the manufacturing process of sodium silicate, which constitutes a high-temperature reaction between silica sand and sodium hydroxide solution. In the case of precursors, around 23% of the GWP in pure GGBS-based AAC is caused by the use of GGBS. Fly ash, on the other hand, is treated as a waste material and makes no contribution to GWP. Besides, there is a rising tendency for the contribution of transport to GWP as GGBS consumption rises, while the contribution is small in the case of PC concrete. This is owing to the procurement of GGBS from a distant source and PC from a nearby supplier. However, the effects of transportation can always be mitigated if the raw materials are sourced from a nearby location.

Comparative study reveals that, across nine out of ten impact categories, PC is the factor that has the greatest influence on environmental deterioration. However, in the case of TEP, the most detrimental effects have resulted from the transfer of raw materials to the manufacturing facility (by a factor of 50–60%). This is due to the exposure of dangerous contaminants such as copper, zinc, lead, nickel, and cadmium to air.

Another significant impact category to be considered is ODP, which corresponds to a decline in stratospheric ozone concentration over an infinite time horizon [77]. PC and sodium silicate are the major contributors to ODP, with PC contributing 60% and sodium silicate around 40–45%. A significant component in the manufacturing of sodium silicate is sodium hydroxide. The chlor-alkali process, which is used to produce sodium hydroxide, uses carbon tetrachloride to liquefy and purify chlorine as well as scrub nitrogen trichloride [86]. This, therefore, turns into a significant factor in the depletion of the ozone layer.

Figure 5 depicts the normalized impact scores of the plain concrete mixtures under each impact category. The normalization and weighting set, World (2010) H, is employed to normalize the midpoint impact categories. The normalization coefficients applied to convert ReCiPe midpoint impact results are specified in Table 6. As a result of being

normalized, these scores provide a straightforward approach to compare different mix designs and determine the most environmentally sustainable one. The findings show that PC concrete has the worst effects on GWP, TAP, EOFP, FEP, and FRS, which is primarily related to PC manufacturing. The S100 mixture has the maximum impact for all other impact categories, mostly due to sodium silicate and an increased use of GGBS. It is evident that a higher GGBS addition leads to greater environmental damage, which is attributed to the discharge of toxic trace metals to groundwater during the processing of blast furnace slag [87]. The conveyance of GGBS from a remote source is yet another factor contributing to potential pollution. In general, the detrimental effects of AAC mixtures are mostly attributable to the chlor-alkali process used to produce sodium silicate, which is a key component of the alkaline-activator. Future research can focus on the valorization of aluminosilicate-rich waste materials with a high Si/Al ratio, necessitating smaller amounts of sodium silicate [12]. Alternatively, silica-rich waste by-products such rice husk ash can be investigated as potential replacements for sodium silicate manufacturing [88].

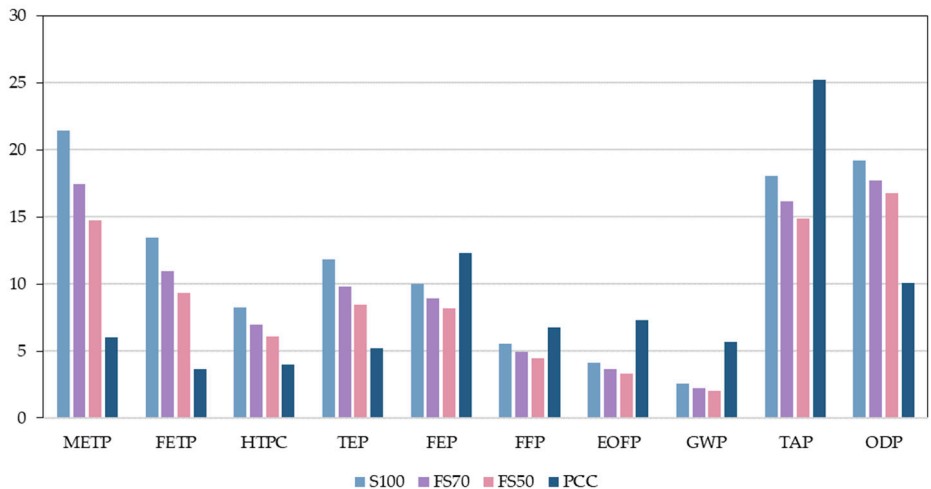

**Figure 5.** Normalized scores of AAC and PC concrete on midpoint impact categories.

**Table 6.** Normalization factors for midpoint impact categories.

| Impact Category | Unit | World (2010) H Normalization Factors |
|---|---|---|
| Climate change | kg $CO_2$-eq to air | $4.18 \times 10^{13}$ |
| Ozone Depletion | kg CFC-11-eq to air | $2.10 \times 10^{8}$ |
| Photochemical oxidant formation: terrestrial ecosystems | kg NOx-eq to air | $3.51 \times 10^{11}$ |
| Terrestrial Acidification | kg $SO_2$-eq to air | $3.18 \times 10^{11}$ |
| Freshwater Eutrophication | kg P-eq to freshwater | $3.77 \times 10^{1}$ |
| Human Toxicity (Cancer) | kg 1,4 DCB-eq to urban air | $1.20 \times 10^{12}$ |
| Terrestrial Ecotoxicity | kg 1,4 DCB-eq to industrial soil | $3.72 \times 10^{10}$ |
| Freshwater Ecotoxicity | kg 1,4 DCB-eq to freshwater | $2.94 \times 10^{10}$ |
| Marine Ecotoxicity | kg 1,4 DCB-eq to marine water | $2.85 \times 10^{10}$ |
| Fossil Resource Scarcity | kg oil-eq | $7.78 \times 10^{12}$ |

Note: Data extracted from [89].

While the midpoint assessment analyzes the potential environmental implications of each mix design, the endpoint assessment determines the environmental impacts at the end of the cause–effect chain. Since numerous environmental impact pathways eventually result in damage to human health, damage to ecosystems, or resource depletion, the endpoint assessment can help draw a better conclusion on the most environmentally sustainable mixture.

### 3.1.2. Endpoint Assessment

Figure 6 evidently demonstrates PC concrete to have the greatest impact on human health and ecosystem quality and the S-100 mixture to have a slightly greater impact than PC on resource scarcity. In comparison to all four mixtures, FS50 has the least impact on all three areas of protection. This is mainly due to the higher replacement levels of GGBS with fly ash, as it is a waste material free from any environmental burden. In categories of damage to human health and ecosystem, the S100 mixture has the most impacts next to PC concrete. This is owing to the usage of sodium silicate as a key component in the alkaline-activator and the use of GGBS as the sole precursor, which causes detrimental effects to the ecosystem and human health, as predominantly observed in the midpoint assessment. However, the environmental impacts of the S100 mixture are 46% and 25% lower than those of PC concrete, respectively, for ecosystem quality and human health. However, there is a negligible increase (6%) in the impacts of the S100 mixture with respect to PC concrete on resource scarcity. This increase is associated with the iron-ore mining operation required to obtain slag as the by-product.

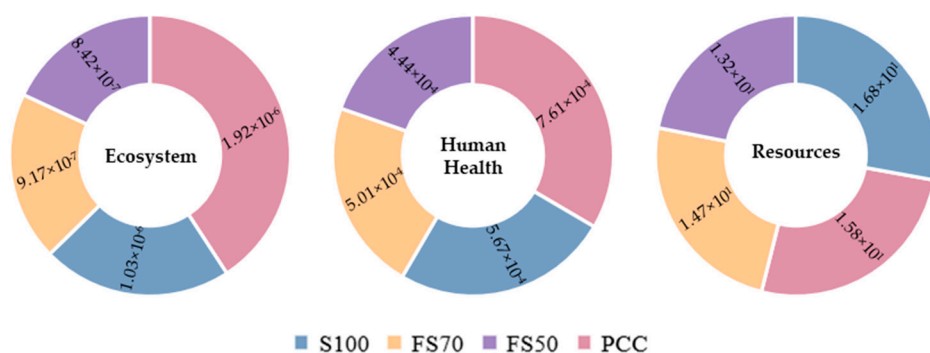

**Figure 6.** Endpoint impacts of AAC and PC concrete.

Based on the outcomes of the endpoint assessment, it can be concluded that AAC mixtures are more sustainable for the ecosystem compared to PC concrete. The mixture with the lowest environment impact among the AAC mixtures is the mix with 50% fly ash and 50% GGBS (FS50). Hence, further analysis on FRAAC is conducted for the FS50 mixture.

### 3.2. Fiber Reinforced Alkali Activated Concrete

### 3.2.1. Midpoint Assessment

The impact assessment results of FRAAC for ten midpoint categories are shown in Figure 7. Steel, polypropylene, and glass fibers are the 3 different fibers that are added to the FS50 mix at a constant volume fraction of 0.3%. The mix design for FRAACs is given in Table 2. Hence, the only distinguishing factors in the composition of FRAAC are the type of fibers used and the transport distance to the production site (in this case, our laboratory). In eight out of ten impact categories, glass fibers have the highest impact relative to other fiber types. High temperatures (~1700 °C) are required for the manufacture of glass in order to melt silica sand and other raw ingredients. In addition to this, soda ash is used to lower the melting temperatures. As a result, the melting process consumes a lot of energy and releases greenhouse gases such as $CO_2$ and $N_2O$, which contribute to global warming. Among all the impact categories, the contribution of glass fibers is more towards ozone depletion (21%). This is attributed to the synthesis of compounds containing chlorine or bromine groups during the manufacture of glass, including chlorofluorocarbons (CFCs), hydrochlorofluorocarbons (HCFC), and other chemicals [90]. Furthermore, the decomposition of sulphate in the batch materials for glass fibers results in the generation of sulphur-di-oxide ($SO_2$), which is the primary cause for acidification. Besides the manufacturing process of glass fibers, the conveyance of these fibers from a distant location also contributes slightly to the environmental impacts.

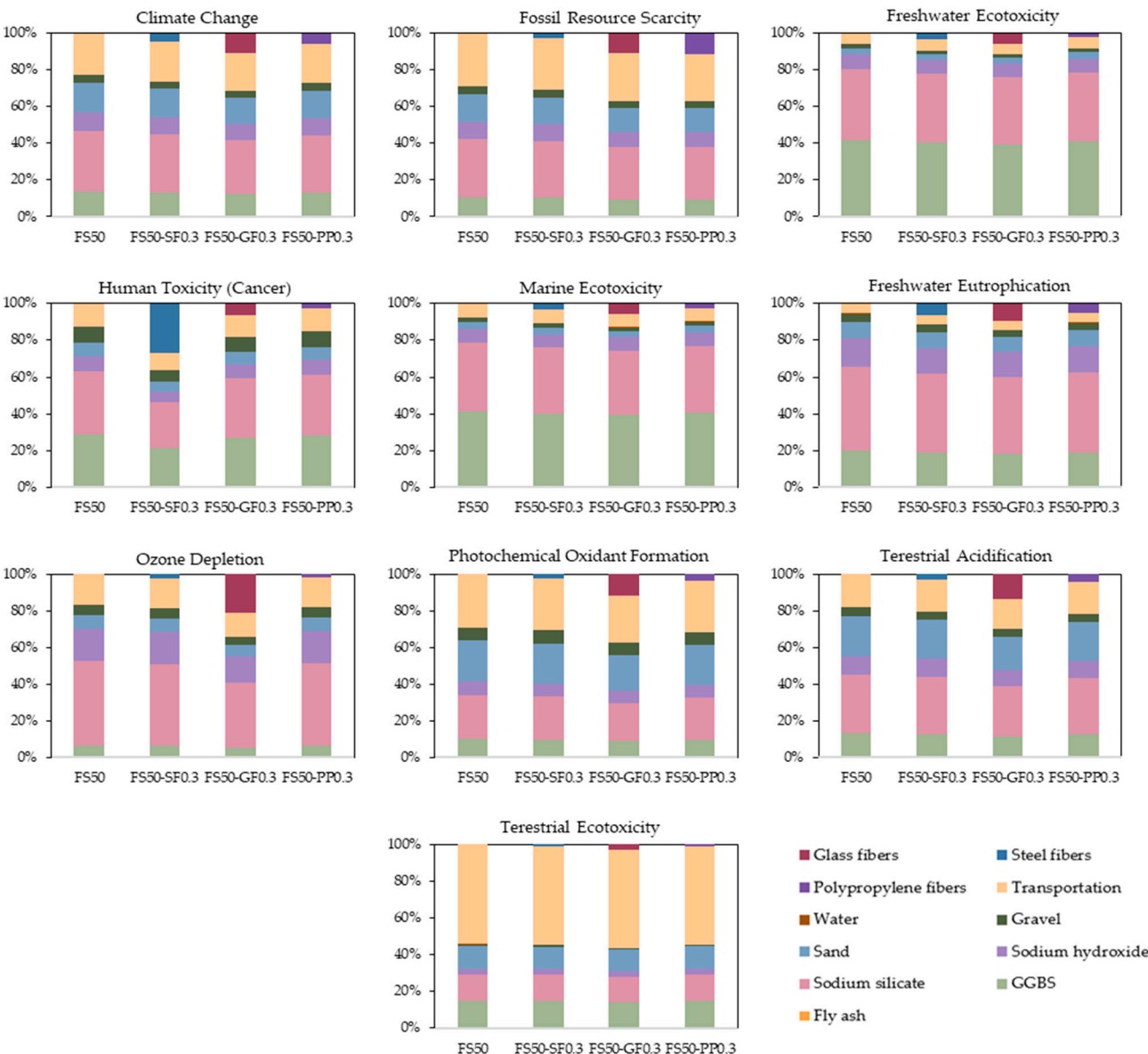

**Figure 7.** Process contribution (%) of FRAAC on midpoint impact categories.

Steel fibers have the highest potential on human cancer rates than any other fiber under consideration. Large amounts of coke are needed for the production of steel, which is highly detrimental to the environment. Air pollution from coke furnaces includes carcinogenic chemicals such as naphthalene. In addition to being extremely hazardous, waste water from the coking process also contains cyanide, sulphides, ammonium, and ammonia in addition to a number of organic chemicals that are known to cause cancer. In general, the environmental effects of using polypropylene fibers are the lowest. Despite this fact, polypropylene is a thermoplastic fiber that is mainly composed of hydrocarbons and is derived from fossil fuels. Hence, fossil fuel is continuously excavated and a lot of energy is consumed in the process, depleting the fossil fuel reservoir and energy sources, to satisfy the high demand for this fiber.

The normalized scores of FRAACs on the midpoint impact categories are presented in Figure 8. Nine out of ten impact category assessments indicate that AAC with glass fibers has the highest impact when compared to other FRAACs. These findings contradict the earlier research [13], which shows glass fibers to have the least impact compared to steel and polypropylene fibers. The variation in results is attributed to the use of a varying proportion of fibers in AAC and the use of a different database for the impact analysis.

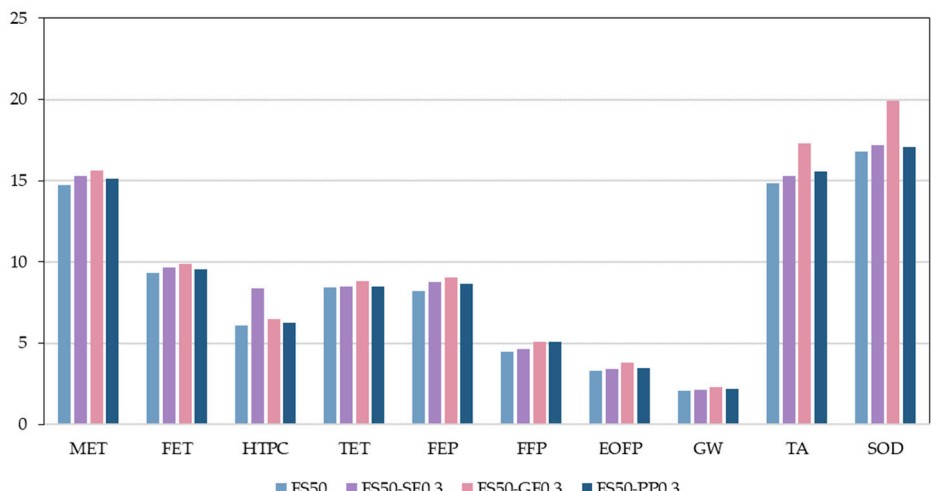

**Figure 8.** Normalization scores of FRAAC on midpoint impact categories.

The impacts of polypropylene fiber reinforcements (FS50–PP0.3) are also comparable with glass fibers (FS50–GF0.3) in MET, FET, HTPC, TET, FEP, FFP, EOFP, and GWP. FS50–GF0.3 show very high impacts in TA and SOD, owing to the emission of CFCs, HCFC, and other chemicals [90]. However, steel fibers have the maximum impact on HTPC owing to the release of harmful toxins during the manufacture of coke, which is largely used for the production of steel.

### 3.2.2. Endpoint Assessment

The endpoint assessment results (Figure 9) reveal that the impacts of all FRAACs are comparable and there is a considerable rise in the impacts of FRAACs in comparison with plain AAC. FS50–GF0.3 has the maximum impact on human health and ecosystem quality, while FS50–PP0.3 has the highest impact on resource scarcity. This is attributed to the high melting temperatures required for the manufacture of glass fibers and high demand for fossil fuels for the production of polypropylene fibers. Among the three different fibers used for FRAAC, steel has the least impact on ecosystem quality and scarcity of resources. However, in the case of human health, steel fibers are only the second-best choice, next to polypropylene fibers. Compared to FS50–PP0.3, FS50–SF0.3 has a marginal increase of 4.3% in affecting human health. Based on the endpoint assessment results, glass fibers are the least environmentally sustainable fibers to be incorporated in AAC, whereas steel fibers have the least negative impacts.

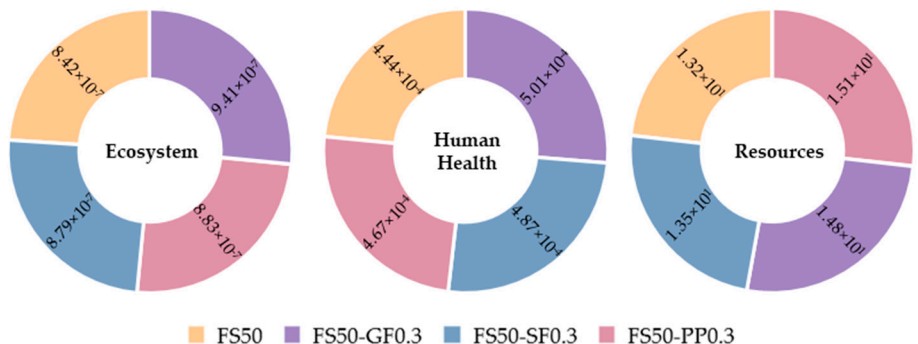

**Figure 9.** Endpoint impacts of FRAAC.

### 3.3. Comparison with ILCD Recommended Impact Assessment Methods

For several impact categories, the ILCD suggests default techniques for impact assessment. In order to validate the reliability of the ReCiPe midpoint method, the present study employs several other LCIA methods, as suggested in the ILCD handbook [82].

Consequently, these LCIA results are compared with those of the ReCiPe 2016 midpoint (H) v1.1 method. The LCIA methods selected on the basis of the recommendations of ILCD handbook are: (i) IPCC for GWP, (ii) USEtox v1.0 for FETP and HTPC, and (iii) CML for ODP.

Figure 10 compares the impact assessment findings obtained using the ReCiPe midpoint and the ILCD suggested methods. Since the impacts are obtained in different units for the ReCiPe, USEtox, and CML methods, a comparison is performed based on the normalized contribution analysis. The impact assessment results employing the ReCiPe midpoint method match closely with those of ILCD recommended approaches for the impact categories of climate change and fossil resource scarcity. This similarity may be traced back to the underlying models employed in the LCIA methods. The IPCC Baseline Model of 100 Years is the recommended approach to climate change by ILCD. Likewise, IPCC [91] and [92] are used as the reference models for the ReCiPe midpoint approach for computing GWP. The primary objective of ReCiPe is to build an updated method that combines Eco-Indicator 99 and CML [80], which is recognized as the likely cause for the similarities between the impact assessment of ReCiPe and CML for fossil resource scarcity.

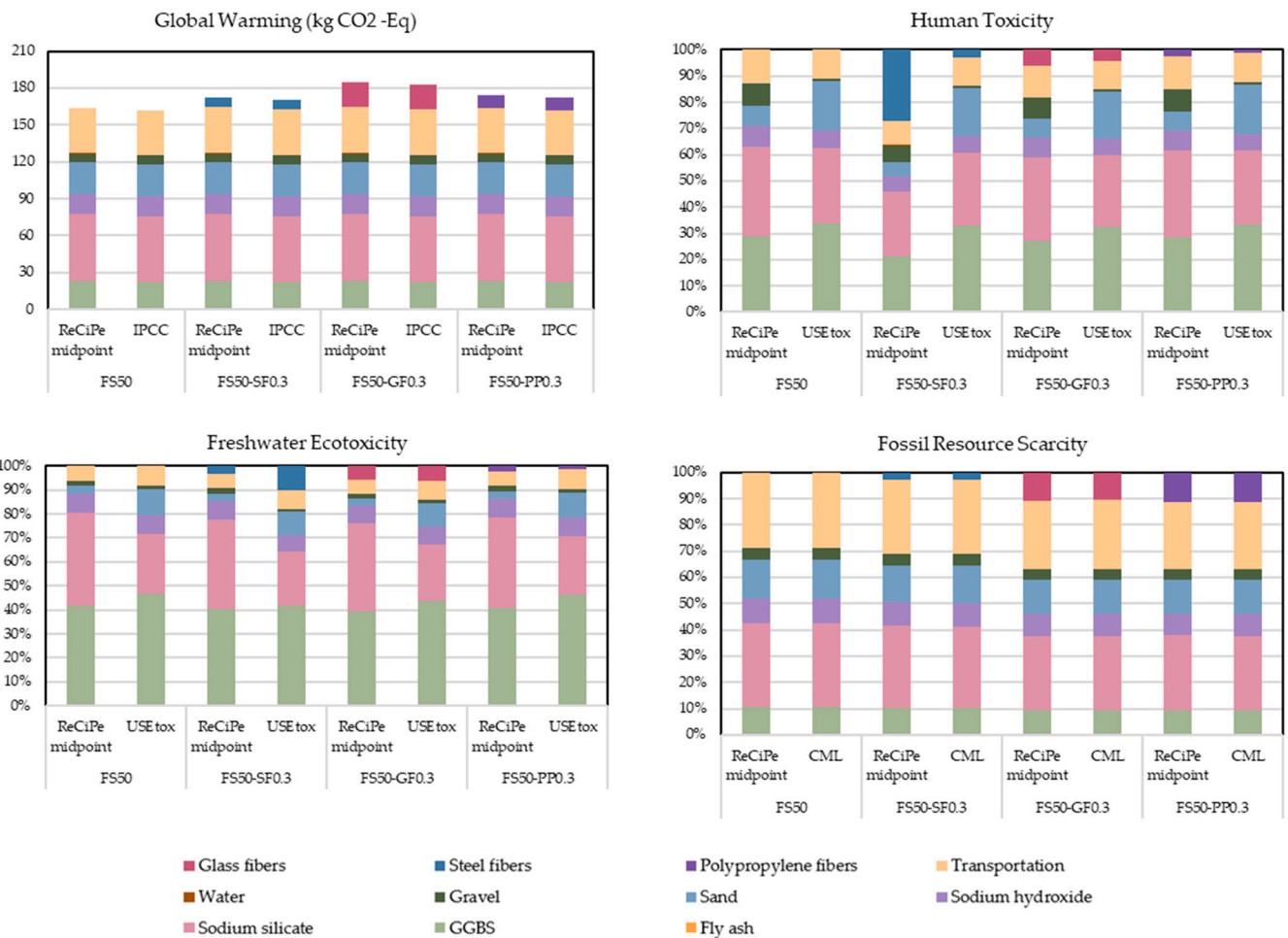

**Figure 10.** Comparison of ReCiPe midpoint assessment results with ILCD recommended methods.

On the contrary, when toxicity potentials are calculated using the USEtox method and ReCiPe midpoint technique, there is substantial difference in the impact results. This is attributed to the fact that both LCIA approaches employ different reference models. The Universal System for Evaluation of Substances (USES-LCA 2.0) serves as the foundation for calculating toxicity potentials in ReCiPe. On the other hand, USEtox bases its calculations on the existing reference model [93]. The USEtox model [93] is not used for implemen-

tation in ReCiPe 2016 because it lacks characterization factors for terrestrial and marine toxicity [79]. Another practical justification for favoring USES-LCA to USEtox is that the latter does not readily offer the opportunity to evaluate the impact of value decisions on the characterization factors, such as the option to create time horizon dependent characterization factors [79]. Furthermore, the USEtox model lacks a characterization factor for chlorine emissions.

### 3.4. Cost Analysis

It is crucial for FRAAC to be commercially feasible in order to support its commercial use. To determine the economic viability of FRAAC in comparison to conventional PC concrete and plain AAC mixtures, a comparative economic analysis has been conducted (Figure 11).

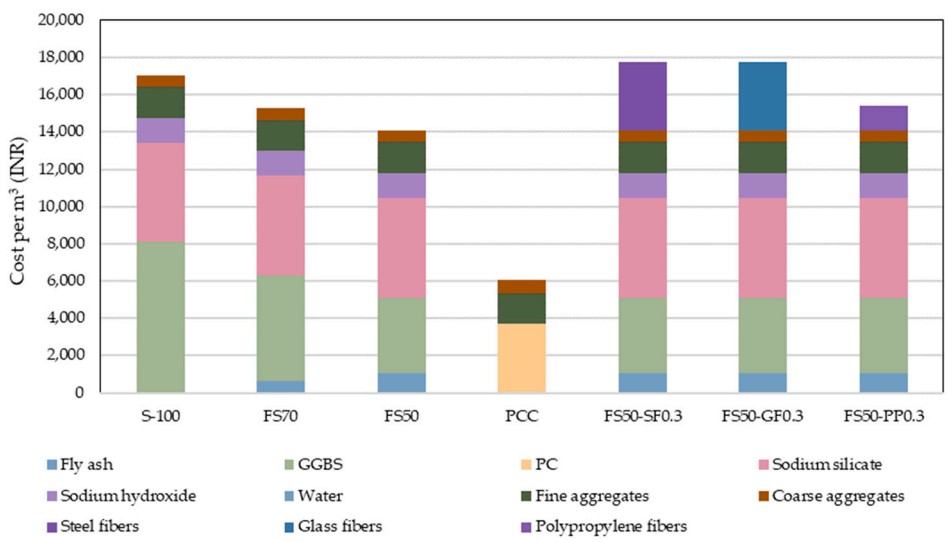

**Figure 11.** Cost analysis of AAC, FRAAC, and PC concrete.

According to Figure 9, the cost of producing 1 m³ of all AAC mixtures is significantly greater than that of traditional PC concrete. Producing 1 m³ of plain AAC concrete ranges from INR 14,000 to INR 17,000, whereas that of PC concrete is INR 6000. The high cost of producing AAC mixtures is attributed to the use of sodium silicate, which is a major component of the alkaline-activator in AAC mixes. Nevertheless, sodium silicate can be replaced with other silica-rich waste products to reduce the price of the alkaline-activator. Another significant observation is that the usage of fly ash results in a cost reduction compared to GGBS. A higher replacement of GGBS with fly ash (FS50) leads to a reduction in cost by 21% compared to AAC produced with GGBS as the sole binder. However, compared to PC concrete, the cost of plain AAC mixtures is about 132–181% higher.

In the case of FRAACs, adding fibers to the FS50 mixture has resulted in a 9.5% to 26% increase in cost. Among the three evaluated fiber additions, the inclusion of polypropylene fibers had the lowest cost impact (9.5%), while steel fibers have the highest impact. Despite being less expensive in terms of weight, steel fibers are added in greater quantities to meet the required 0.3% volume dosage, which is attributed to the cause of the increased expense of employing steel fibers in the manufacture of FRAAC. Nonetheless, the economic impacts of adding glass fibers are nearly identical to those of adding steel fibers, with a marginal difference of 1.02%. It can also be observed that polypropylene fiber reinforced AAC (FS50-PP0.3) has a lower cost of production when compared to plain AAC with GGBS as the sole precursor.

The findings from this study demonstrate the higher environmental impact of PC concrete compared to AAC having similar compressive strength. However, in the case of AAC, the use of sodium silicate shows detrimental effects towards the environment.

Among GGBS and fly ash precursors used in AAC, GGBS has higher impacts, whereas fly ash makes no contribution to any of the impact categories as it is treated as a waste material obtained from the industry. As a result, the higher replacement of GGBS with fly ash leads to reduced environmental impacts. The addition of glass fibers in AAC has the highest impact relative to steel and polypropylene fibers. Though AAC and FRAAC outperform PC concrete in terms of environmental sustainability, the cost of producing these mixes is higher compared to PC concrete. Further research into the development of sodium silicate from silica-rich waste materials can help in reducing the cost as well as environmental impacts of AAC.

## 4. Conclusions

In the current study, four different plain concrete mixtures are subjected to a detailed life cycle assessment to determine their environmental impacts on both midpoint and endpoint levels. Based on the investigation, the FS50 mixture is found to be optimal and is considered for further environmental assessment of steel, glass, and polypropylene FRAAC. To determine whether these mixtures are commercially viable, a simple cost analysis is also carried out. The outcomes are outlined as follows:

- Among the plain concrete mixtures, PC concrete is found to have 86% and 34% higher impacts than AAC on ecosystem quality and human health, respectively. Portland Cement is the key contributor to the high environmental impacts of PC concrete.
- The impacts on environment from AAC are mostly attributed to the use of sodium silicate in the activator. In all midpoint impact categories, sodium silicate accounts for 30–50% of the total impact.
- The partial replacement of GGBS with fly ash helps in reducing the environmental impacts of AAC by 7–18%. An AAC mixture with 50% fly ash and 50% of GGBS (FS50) has the least environmental impacts among the plain concrete mixtures.
- Among the FRAAC mixtures, glass fiber addition resulted in higher impacts to the environment, compared to polypropylene and steel fibers.
- The addition of glass fibers at 0.3% volume fraction in the FS50 mixture resulted in 12% and 13% higher impacts on the quality of the ecosystem and human health, respectively, when compared to FS50 without fibers.
- The cost analysis indicated that AAC has at least a 132% increase in costs when compared to conventional PC concrete. The increase in the cost of AAC is mainly attributed to the use of sodium silicate in the alkaline-activator. In FRAACs, the addition of 0.3% of polypropylene fibers resulted in the lowest impact on production cost.

The current study provides a comprehensive analysis of the environmental effects of FRAAC throughout its life cycle. The concrete industry and policymakers would find a feasible and sustainable alternative to PC concrete through the assessment of the environmental impacts of various concrete mixtures at midpoint and endpoint damage levels. The authors agree that the findings from the present study can be used to provide recommendations for the practical use of FRAAC at locations having similar climatic and economic scenarios as the Indian subcontinent.

**Author Contributions:** Conceptualization, P.G.C. and A.K.; methodology, P.G.C.; software, P.G.C.; validation, P.G.C.; formal analysis, P.G.C. and A.K.; investigation, P.G.C.; data curation, P.G.C. and A.K.; writing—original draft preparation, P.G.C. and A.K.; writing—review and editing, P.K.D.M.; supervision, A.K. All authors have read and agreed to the published version of the manuscript.

**Funding:** This research received no external funding.

**Data Availability Statement:** All data pertaining to the paper is mentioned in the manuscript. Additional information will be available upon request.

**Conflicts of Interest:** The authors declare no conflict of interest.

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
