# Peer review of "Environmental Impact Analysis of Alkali-Activated Concrete with Fiber Reinforcement"

_infrastructures, doi:10.3390/infrastructures8040068_

Round 1
Reviewer 1 Report
This study provides a comprehensive environmental investigation of AA materials with various types of fibers. In my opinion, the article deserves publication in “INFRASTRUCTURES”. I have minor concerns and comments to address the deficiency of this research.
1) Why is the fiber content not varying. What is the logical reason behind the selection of this dosage of fibers.
2) In the abstract include the level of fiber volumes.
3) Include the results on the engineering properties of AAMs with different fibers.
4) Quality of all figures must be original and refined.
5) Too many unnecessary referencing must be avoided for a simple piece of information.
6) Research flow should also be included in the methodology.
7) In figure 9 the unit of vol. is not correct.
8) Include the unit impact of materials in the inventory to facilitate the readers and future research.
9) Include a map of area showing the location/points of different raw materials considering the case study that was adopted in this study.
10) Some writing mistakes in the introduction needs to be removed.
Reviewer 2 Report
The paper investigates the use of fiber reinforcement on the alkali activated fly ash concrete. The authors conducted different kind of tests. The paper needs to be enhaced before publish.
Abstract should include important results of the studies.
The following studies can be included in the introduction: Geopolymer concrete with high strength, workability and setting time using recycled steel wires and basalt powder; Production of Perlite-Based-Aerated Geopolymer using Hydrogen Peroxide as Eco-friendly Material for Energy-Efficient Buildings
Add a flow chart for experiments
Add photos for materials used in this study.
The authors did not mention which type of steel fiber utilized. The authors are engouraged to add a parahrtaph related to steel fibers. The following can be utilzied for this purpose: improvement in bending performance of reinforced concrete beams produced with waste lathe scraps; performance assessment of fiber-reinforced concrete produced with waste lathe fibers; performance evaluation of fiber-reinforced concretes produced with steel fibers extracted from waste tire; investigation on improvement in shear performance of reinforced-concrete beams produced with recycled steel wires from waste tires; experimental and numerical investigations of steel fiber reinforced concrete dapped-end purlins
Add discussion for setting time
Add discussion for workability.
Add photos for test setup
Reviewer 3 Report
The submitted manuscript in infrastructures-2300801 lacks the following:
1- The abstract is very poorly written and it sounds more like an introduction rather than an abstract.
2- provide pictures of the binders in the lab.
3- provide pictures of the concrete mix and the samples used.
4- In section 2- 3 provide more details about the life cycle assessment method.
5- Figure 1 is not clear to the reader.
6- Provide more details in conclusions.
Reviewer 4 Report
- Even if the study takes into account data taken from regions in India, it can also be adapted at the global level, the conclusions being probably largely similar;
- It would be interesting to do an analysis regarding the evolution over time of the performance of alkaline activated concretes in order to follow the trend of research in the field.
Round 2
Reviewer 2 Report
Accept